

# OpenDrift v1.0: a generic framework for trajectory modeling

Knut-Frode Dagestad [1], Johannes Röhrs [1], Øyvind Breivik [1,3], and Bjørn Ådlandsvik [2]

[1]Norwegian Meteorological Institute, Bergen, Norway
[2]Institute of Marine Research, Bergen, Norway
[3]Geophysical Institute, University of Bergen, Norway

*Correspondence to:* Knut-Frode Dagestad (knutfd@met.no)

**Abstract.** OpenDrift is an open-source Python-based framework for Lagrangian particle modeling under development at the Norwegian Meteorological Institute with contributions from the wider scientific community. The framework is highly generic and modular, and is designed to be used for any type of drift calculations in the ocean or atmosphere. A specific module within the OpenDrift

framework corresponds to a Lagrangian particle model in the traditional sense. A number of modules have already been developed, including an oil drift model, a stochastic search and rescue model, a pelagic egg model, and a basic module for atmospheric drift. The framework allows for the ingestion of an unspecified number of forcing fields (scalar and vectorial) from various sources, including Eulerian ocean, atmosphere and wave models, but also measurements or subjective estimates of

the same variables. A basic backtracking mechanism is inherent, using sign reversal of the total displacement vector and negative time stepping. OpenDrift is fast and simple to set up and use on Linux, Mac and Windows environments, and can be used with minimal or no Python experience. It is designed for flexibility, and researchers may easily adapt or write modules for their specific purpose. OpenDrift is also designed for performance, and simulations with millions of particles may

be performed on a laptop. Further, OpenDrift is designed for robustness, and is in daily operational use for emergency preparedness modeling (oil drift, search and rescue and drifting ships) at the Norwegian Meteorological Institute.

## 1   Introduction

Lagrangian trajectory models are used to predict the pathways and transformations of various types
of objects and substances drifting in the ocean or in the atmosphere. There are many practical and academic applications, including prediction of:

- oil drift and weathering to aid mitigation and cleanup operations (Jones et al., 2016)



- drifting objects for search and rescue (Breivik and Allen, 2008; Breivik et al., 2011, 2013)

- ichtyoplankton transport (fish eggs and larvae) for stock assessments (Röhrs et al., 2014)

- microplastics suspended in the ocean (van Sebille et al., 2012, 2015)

Table 1 lists some commonly used trajectory models and their applications. Additionally, many individual researchers or research groups have been developing trajectory model codes for in-house use, without publishing (or naming) a software code.

Lagrangian tools fall in two broad categories, either the trajectories are computed along with the
velocity fields as part of the ocean or atmospheric circulation model (e.g. so-called floats in the regional ocean modelling system (ROMS), Shchepetkin and McWilliams 2005). This is known as *online* trajectory computations, and has the advantage that no separate model is needed. Alternatively, the trajectories can be computed *offline* after completion of the Eulerian model simulation(s). This is the approach taken for OpenDrift, and is also necessary for a generic framework as the tra-
jectories depend in many cases on forcing from a range of fields stemming from more than just one Eulerian model. This is e.g. the case for oil drift and search and rescue models, which both require wind as well as currents (and wave forcing in the case of oil drift) to properly account for the advection and transformation of the particles. For such emergency preparedness purposes, offline models are the only option fast enough to meet the requirements of operational agencies. Another advantage
of offline models is that modifications (sensitivity tests) of the drift algorithms may be tested quickly without needing to rerun the full Eulerian model.

Existing trajectory models are in most cases tied to a specific application, and may not be applied to other drift applications without compromising quality or flexibility. In many cases, trajectory models are also tied to a specific Eulerian model, or even a particular institutional ocean model setup,
limiting usability for other institutes or researchers. Often, it is also required that Eulerian forcing data must be in a specific file format. This raises the time and effort needed to set up a trajectory model. Further, in an operational setup, the need to convert large files to another format increases both the complexity and computational costs of the processing chain, compromising robustness.

The OpenDrift framework has been designed to perform all tasks which are common to trajectory
models, whether oceanic or atmospheric. In short, the main task is to obtain forcing data from various sources, and to use this information to move (propagate) the elements in space, while potentially transforming other element properties, such as evaporation of oil, or growth of larvae. In addition, common functionality includes mechanisms for configuration of simulations, seeding of elements, exporting output to file, and tools to visualise and analyse the output. Additionally, several design
requirements have been imposed on the development of OpenDrift: (1) platform independence and ease of instalment and use; (2) simple and rapid implementation of any purpose-specific processes, yet flexibility to support unforeseen needs; (3) forcing data from any type of source shall be supported, including Eulerian ocean, atmosphere or wave models (typically NetCDF or GRIB files), *in*





*situ* measurements, vector datasets (e.g. GSHHS coastlines), or analytical fields for conceptual stud-

ies; (4) it must run fast, even with a large number of elements; (5) simulations forward and backward
in time; (6) robustness for operational use.

## 2   Software design

To meet the requirements listed above, a simple and flexible object oriented data model has been de-

signed, based on two main classes. One class ("Reader") is dedicated to obtaining forcing data from
external sources, as described in Sec 2.1. A generic class for a trajectory model instance ("Base-
Model") is described in Sec 2.2. This class contains functionality which is common to all drift
models, whereas advection (propagation) and transformation of elements is left for purpose-specific
subclasses.

### 2.1   Reader class

The class *Reader* obtains forcing data (e.g. wind, waves and currents) from any possible source, and
provides this to any OpenDrift model through a common interface. To avoid duplication of code, a
parent class "BaseReader" contains functionality which is common to all Readers, whereas specific
subclasses take care of only the tasks which are specific to a particular source of data, e.g. how
to decode and interpret a particular file format. Two methods must be implemented by any Reader

subclass: 1) a constructor method which initialises a Reader object, and 2) a method to retrieve data
for given variables, position and time. The constructor (__init__ in Python) can take any arguments
as implemented by the specific Reader class, but typical is to provide a filename or URL from which
data shall be obtained by this Reader. The following Python commands initialise a reader of type

NetCDF_CF_generic to obtain data from a file "ocean_model_output.nc".



| Name | Reference / URL | Main application |
|------|-----------------|------------------|
| BSHDmod | Dick and Soetje (1990) | Oil |
| CIS iceberg model | Kubat et al. (2007) | Icebergs |
| CLaMS | McKenna et al. (2002) | Atmospheric chemistry |
| EMEP | Simpson et al. (2012) | Air pollution |
| FLEXPART, FLEXSTRA | www.flexpart.eu Stohl et al. (1995) | Nuclear, air pollution |
| HYSPLIT | Stein et al. (2015) | Atmospheric transport |
| Ladim | Ådlandsvik and Sundby (1994) | Plankton transport |
| LAGRANTO | Wernli and Davies (1997), Sprenger and Wernli (2015) | Meteorology |
| LAGRANTO.ocean | Schemm et al. (2017) | Water mass properties |
| Leeway | Breivik and Allen (2008), Allen and Plourde (1999) | Search and rescue |
| LTRANS | Schlag and North (2012) | Plankton (including larvae) |
| MEDSLIK, MEDSLIK-II | De Dominicis et al. (2013), Lardner et al. (1998) | Oil |
| MIKE | www.mikepoweredbydhi.com | Oceanic, generic |
| MOHID | www.mohid.com | Oil, sediments, water quality |
| MOTHY | Daniel (1996) | Oil, drifting objects |
| OD3D | Wettre et al. (2001) | Oil |
| OILMAP, SIMAP, CHEMMAP, MUDMAP, SARMAP | www.asascience.com | Oil, sediments, chemical, search and rescue |
| OILTOX | Brovchenko et al. (2003) | Oil |
| OILTRANS | Berry et al. (2012) | Oil |
| OSCAR | www.sintef.no/en/software/oscar | Oil |
| OSERIT | oserit.mumm.ac.be | Oil, chemicals |
| PARCELS | github.com/OceanPARCELS/parcels | Ocean, generic |
| POSEIDON-OSM | osm.hcmr.gr | Oil |
| PyGNOME/GNOME | gnome.orr.noaa.gov | Oil, generic |
| SeaTrackWeb, PADM | stw.smhi.se | Oil, chemicals |
| SNAP | Bartnicki et al. (2016) | Atmospheric nuclear transport |
| STILT | www.stilt-model.org | Atmospheric trace gases |
| THREETOX | Margvelashvily et al. (1997) | Nuclear ocean transport |
| TRACMASS | Döös et al. (2013) | Ocean and atmosphere, generic |
| VOS | en.ferhri.org | Oil |

**Table 1.** Some existing trajectory models for various oceanic and atmospheric applications.

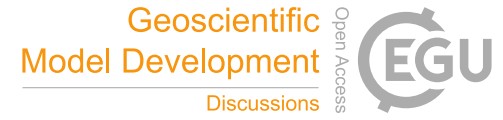

```
>>> from opendrift.readers import reader_netCDF_CF_generic
>>> r = reader_netCDF_CF_generic.Reader("ocean_model_output.nc")
```

The initialisation typically includes opening and reading metadata from a given file or URL to check
which variables are available, and the coverage in time and space. The contents can be inspected by
printing the object:

```
>>> print r
Projection:
  +proj=stere +lat_0=90 +lon_0=70 +lat_ts=60 +units=m +a=6.371e+06 +e=0 +no_defs
Coverage: [m]
  xmin: -2952800.000000   xmax: -2712800.000000   step: 800   numx: 301
  ymin: -1384000.000000   ymax: -1224000.000000   step: 800   numy: 201
  Corners (lon, lat):
    ( 2.52,  59.90) ( 4.28,  61.89)
    ( 5.11,  59.32) ( 7.03,  61.26)
Vertical levels [m]:
  [0.]
Available time range:
  start: 2015-11-16 00:00:00   end: 2015-11-18 18:00:00   step: 1:00:00
    67 times (0 missing)
Variables:
  x_sea_water_velocity
  y_sea_water_velocity
```

This above example shows that the created Reader object can provide ocean surface current on a grid
with 800 m pixel size in polar stereographic coordinates, at hourly time resolution.

To allow for generic coupling of any OpenDrift model with any Reader, a naming convention for
variables is necessary. By convention, the commonly used CF naming convention (`cfconventions.org`)
should be used whenever possible. Thus if the data source is not already following this convention
(e.g. a GRIB file), the Reader should map the variable names to corresponding CF *standard_name*.

    The given Reader class must also have implemented a specific instance of the method *get_variables*
which is called to return data:

```
>>> data = r.get_variables(["x_wind", "y_wind"], x, y, z, time)
```

The horizontal coordinates (`x, y`) correspond to the native projection of the Reader, which is polar
stereographic in the given example. The task of transforming from one coordinate system to another
(including the rotation of vectors) is performed by common methods from the parent class, based
on the widely used proj.4 library (`proj4.org`), through its Python interface pyproj. This allows





OpenDrift to combine input data from any coordinate systems, whilst keeping the implementation of new Reader classes as minimalistic and clean as possible. Further, this centralisation of code also facilitates optimisation for both performance and robustness. The vertical coordinate ($z$) is by convention always in meters, zero at the air/water surface and positive upwards. Thus Readers providing data from sources with other vertical coordinate systems (e.g. topography following coordinates or pressure/density levels) must take care of transforming this to meters before data is returned. This is e.g. done by an existing reader supporting native output from the ROMS ocean model. The variable *time* is consistently handled as Python *datetime* objects within OpenDrift, with any time intervals as *timedelta* objects. Readers also share some common convenience methods, such as plotting of geographical coverage.

Readers are, however, normally not called directly by the user or from specific OpenDrift instances (models), but rather implicitly from the parent BaseModel class (see Sec 2.2). A caching mechanism is implemented to minimize the amount of data to be read, which is a key to improving performance. Input data from numerical models are normally provided on a 3D spatial grid $(x, y, z)$ at discrete time steps, which is often larger than the time steps used internally by OpenDrift models. When data is requested for a given set of element positions at a given time, OpenDrift requests from the Readers 3D-blocks of data from the time before and after the given time. These 3D-blocks encompass the elements tightly, except for a buffer on each side which is large enough so that elements will stay within the coverage during the time step of the Reader. After 3D blocks of data are provided by the Reader, interpolators are generated, and then reused to interpolate the same data blocks onto the element positions successively at each internal calculation time step, until the calculation time step reaches the latter Reader (model) time step. At this point, a 3D block for the subsequent model time step is requested, and a new interpolator is generated. Due to this very economical access of remote data, simulations with OpenDrift are almost as fast when obtaining data from remote Thredds servers, as when reading the same data from a local file. The interpolator mechanism is also modularised by a dedicated class in OpenDrift, allowing independent development and optimisation.

Whereas obtaining forcing data from 3D Eulerian models is the most common in practice, Readers may obtain data from any other possible source. One example is to read a time series from an ASCII file of observations, e.g. from a buoy or a weather station. Another example is to calculate forcing data according to some analytical function. One such example is included in the code repository, providing ocean current vectors according to a "perfect circular eddy" with centre coordinates as given to its constructor. Such analytical forcing data fields are useful for e.g. testing the accuracy of forward propagation schemes, as discussed below.

## 2.2 BaseModel class

Functionality which is common to any trajectory model is described in a main class, named Base-Model. This functionality includes the following:



1. A mechanism for configuration of a trajectory model, or a specific simulation. This may include adjusting the resolution of a coastline, or some model specific parameters concerning the movement of the elements. The configuration mechanism of OpenDrift is based upon the ConfigObj package (https://pypi.python.org/pypi/configobj).

2. A generic method to seed elements for a simulation. See Sec 2.3.3 for details.

3. Managing and referencing a set of Readers (Sec 2.1) which are called as needed to obtain forcing data during a simulation. See Sec 2.3.2 for details.

4. Keeping track of the positions and properties of all elements during a simulation, and removing elements scheduled for deactivation. This is stored in 2D arrays with two dimensions, time and particle ID. Thus the trajectory (propagation with time) of a single element or the simulation state (all element positions and properties at a given time) is easily and quickly obtained as vertical or horizontal slices of the array. The history of data may also be written to file, as described in Sec 2.3.6.

5. Finally, the BaseModel class contains the main loop for time-stepping, performing necessary tasks for a simulation in the correct order as described in Sec 2.3.5.

The only part missing is a description of how the elements (e.g. objects or substance) shall be propagated and potentially transformed along their trajectories under the influence of environmental forcing data. Such application specific description is left to subclasses, yielding trajectory model instances as exemplified in Sec 3. These subclasses thus inherit and may reuse any functionality from the BaseModel. The subclasses may also add further functionality as needed, or overload and modify existing functionality. Thus all necessary core functionality is available by convenience, but may be modified for flexibility. In precise terminology, OpenDrift is a framework within which specific trajectory models may be implemented by class inheritance (subclassing). An instance (object) of such a subclass represents a specific trajectory simulation.

### 2.3 Performing a simulation

In this section, we describe and explain the general workflow of a simulation with an OpenDrift model, as illustrated in the flowchart in Fig. 1.

### 2.3.1 Initialisation

The first step is to import a specific OpenDrift model (subclass of BaseModel), and to initialise an instance. The following Python statements import and initialise an instance of the Leeway search and rescue model (Sec 3.1).

```
>>> from opendrift.models.leeway import Leeway
>>> l = Leeway()
```



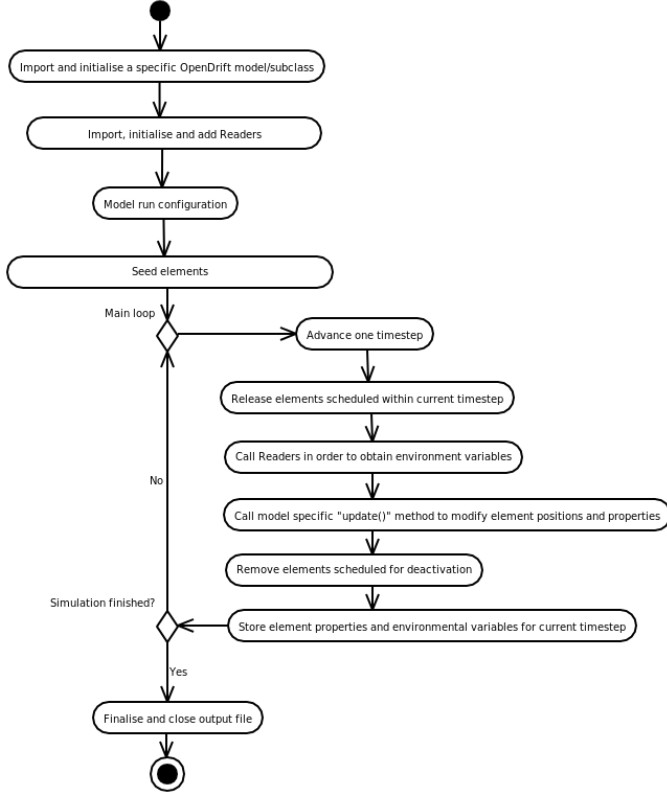

**Figure 1.** Flowchart of an OpenDrift simulation.

### 2.3.2 Adding readers

If a given model requires e.g. ocean current and atmospheric wind as environmental forcing, we need to create and add Reader instances which can provide these variables. Say that we have a Thredds server which can provide ocean currents, and a local GRIB file which contains atmospheric winds, we can create and add these readers to the simulation instance as follows:

```
>>> from opendrift.readers import reader_netCDF_CF_generic
>>> from opendrift.readers import reader_grib
>>> reader_current = reader_netCDF_CF_generic.Reader(
                        'http://thredds.example.com/current.nc')
>>> reader_wind = reader_grib.Reader('winds.grib')
>>> l.add_readers([reader_current, reader_wind])
```

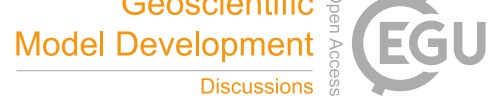



It is also possible to perform a simulation even with no readers added for one or more of the required variables. In this case, constant values may be provided, otherwise reasonable default values will be used, defaulting to zero-values for winds, waves and currents. E.g. in the case of having a 5 day wind forecast, but only a 3 day current forecast, it is still possible to run a 5 day trajectory forecast, where

current will be zero for the last two days.

A key feature of OpenDrift, for both convenience and robustness, is the possibility to provide a priority list of reader for a given set of variables. As an example it is possible to specify that a high resolution ocean model shall be used whenever particles are within coverage in space and time, and reverting to using another model with larger coverage in space and time whenever particles

are outside the time- or spatial domain of the high resolution model. As an important feature for operational setups, the backup readers will also be used if the first choice model (file or URL) should not be available, or if there should be any other problems, such as e.g. corrupt values or files.

### 2.3.3   Seeding of elements

The seeding methods of OpenDrift are very flexible. The simplest case is to seed (initialise) an

element at a given position and time:

```
>>> l.seed_elements(lon=4.0, lat=60.0, time=datetime(2017, 6, 25, 12))
```

Also the number of elements and an uncertainty radius may be provided. Further, both position and time may be provided as two-element vectors to seed elements continuously in space and time from position P1 with uncertainty radius R1 at time T1, to position P2 with uncertainty radius R2 at time

T2. This is a common use case in search and rescue modeling (see Breivik and Allen 2008): a ship is known to have departed from position P1 at time T1 with normally small uncertainty radius R1, and disappeared on the way towards the destination (P2), normally with larger uncertainty in position (R2) and estimated arrival time (T2). Thus, this will track out a 'seeding cone' in space and time. Another common use case is that P1 equals P2, with T2 > T1, e.g. simulating a continuous oil spill

from a leaking well.

Another built-in feature is seeding of elements within polygons. This may e.g. be done by providing vectors of longitude and latitude:

```
>>> l.seed_within_polygon(lon=lonvector, lat=latvector,
                          time=datetime(2017, 6, 25, 12), number=10000)
```

This example will seed 10000 elements evenly within the polygon encompassed by vectors *lonvector* and *latvector*. Based upon this generic polygon seeding method, more specific applications have been developed, see e.g. Sec 3.2.





### 2.3.4 Configuration

OpenDrift modules share several configuration settings which may be adjusted before a simulation,
as well as some settings which are module-specific. All possible settings of a module may be shown
with the command `l.list_configspec()`, of which one example is:

```
drift:scheme [euler] option('euler', 'runge-kutta', default='euler')
```

This shows that the setting `drift:scheme` may have one or two possible values, *'euler'* or *'runge-kutta'*, where the first is the default, and also the present setting as indicated within brackets. A
Runge-Kutta propagation scheme may instead be activated by the command:

```
>>> l.set_config('drift:scheme', 'runge-kutta')
```

The configuration mechanism is based on the widely used ConfigObj package, and allows e.g. exporting to, and importing from, files of the common 'INI'-format.

### 2.3.5 Starting the model run

After initialisation, configuration, adding of readers, and seeding of elements, the model simulation
may be started by calling the method *run*:

```
>>> l.run(duration=timedelta(hours=48), time_step=timedelta(minutes=15),
          outfile='outleeway.nc')
```

This starts the main loop, as shown on the flowchart of Fig. 1. At each time step, forcing data is
obtained by all the readers and interpolated onto the element positions, and the model specific update
method is called to move and/or otherwise update the other element properties (e.g. evaporation of
oil elements, or growth of fish larvae) based on the environmental conditions.

For the above example, the simulation will cover 48 hours, starting the time of the first seeded
elements. The time step of the calculation is given here as 15 minutes. An output time step might be
specified differently, with e.g. output every hour to save memory and disk space.

All instances of OpenDrift can be run in reverse, i.e. backwards from a final destination, by reversing the sign of the advective increment. All spatial increments due to model physics pertinent to the
instance in question are calculated as normal, but the sign of the total increment, $(\Delta x, \Delta y, \Delta z)$, is
reversed and the particles are advected "backwards" over a time step $\Delta t$. All diffusive properties are
kept in the forward sense, meaning that particles will disperse as they propagate backwards in time.
Nonlinear processes such as evaporation of oil or capsizing of vessels, are disabled in backtracking
mode. This simple backtracking scheme is an easy to use alternative to more complicated inverse
methods such as iterative forward trajectory modeling (Breivik et al., 2012b), and is also much less
computationally expensive.

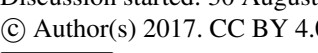



### 2.3.6 Exporting model output

In the above example, the output is saved to a CF-compliant NetCDF file (Trajectory Data specification), which is the default output format of OpenDrift. Both particle positions and any other properties, as well as configuration settings are stored in the file. The simulation may be imported by OpenDrift, or independent software, for subsequent analysis or plotting. Stored output files may also be used as input to a subsequent OpenDrift simulation, allowing for an intermediate step where the particles are subjected to various considerations such as a Bayesian update of their probabilities based on posterior information. Saving data to files is not a requirement, as the output of the simulations is otherwise held in memory for subsequent plotting or analysis, either interactively from within Python shell, or by a script. A number of visualization tools based on the Matplotlib graphics library of Python are included within OpenDrift. Some examples of both generic and module specific plotting methods are illustrated in Sec 3.

## 3 Examples of model instances

### 3.1 Leeway (Search and Rescue)

The OpenDrift Leeway instance (OpenLeeway) is based on the operational search and rescue model of the Norwegian Meteorological Institute (Breivik and Allen, 2008). The model ingests a list of object classes, where each drifting object has specific properties such as downwind and crosswind leeway (the motion due to wind) in a way similar to SAROPS, the operational system used by the US Coast Guard (see Kratzke et al. 2010, and the overview of search and rescue models by Davidson et al. 2009). These properties vary greatly from object to object, and are based on field work (Breivik et al., 2011, 2012a) where specific objects of relevance in search and rescue have been studied. All objects are assumed to be small enough that direct wave scattering forces are insignificant. Furthermore, the Stokes drift (Kenyon, 1969; Breivik et al., 2014, 2016) is inherently part of the leeway obtained from observations. As wind-generated waves have a mean direction closely aligned with the local wind direction it is neither practical nor desirable to disentangle the Stokes drift from the wind drag for leeway simulations.

The uncertainty in the drift properties of a certain object stem from a variety of sources, such as the actual loading of the object (Allen and Plourde, 1999) and from the field studies themselves. Although the definition of leeway is by now well established (Breivik et al., 2013) as

> the motion of the object induced by wind (10 m reference height) and waves relative to the ambient current (between 0.3 and 1.0 m depth),

there is still much uncertainty about the way this motion should actually be measured. Smaller current-measuring devices can now be mounted on much tinier objects than was possible before, and this holds the promise of more precise measurements in the future.





Once an object class has been chosen and the pertinent wind and current forcing fields selected,
the particles are seeded based on the available information. If the particles hit the coast they stick by
default. This can however be relaxed so that particles detach from the coastline if the wind direction
changes.

The OpenLeeway class along with all other subclasses has the option of being run backwards.
This is a convenient feature in cases where for example a debris field is observed and the location
of the accident is sought. Note that this method is fundamentally different from the BAKTRAK
model described by Breivik et al. (2012b) where a large number of particles were seeded in potential
initial locations at various times, and only those that ended up close to the location of the observed
object were kept. This is an iterative procedure which in principle can deal with nonlinearities in
the flow field as well as nonlinear behaviour of the object itself (such as capsizing and swamping).
Although in principle this allows for a more realistic mapping of initial locations, the difficulties
associated with this iterative process means that real-time operations are normally better off with a
simple negative-time integration.

OpenLeeway is used operationally at Norwegian Meteorological Institute, and is also currently
being implemented as the operational search and rescue model for the Joint Rescue Co-ordination
Centres (JRCC) of Norway.

The following lines of Python code illustrate a complete working example of running an Open-
Leeway simulation:





```
from opendrift.readers import reader_netCDF_CF_generic
from opendrift.models.leeway import Leeway
l = Leeway()  # Creating a simulation object
# Wind field
reader_wind = reader_netCDF_CF_generic.Reader(
   'http://thredds.met.no/thredds/dodsC/meps25files/meps_det_pp_2_5km_latest.nc')
# Ocean model data
reader_ocean = reader_netCDF_CF_generic.Reader(
   'http://thredds.met.no/thredds/dodsC/sea/norkyst800m/1h/aggregate_be')
l.add_reader([reader_wind, reader_ocean])
# Seed elements at defined position and time
objType = 26  # Life-raft, no ballast
l.seed_elements(lon=4.5, lat=60.0, radius=1000, number=5000,
                time=datetime(2017,7,1,12), objectType=objType)
# Running the model 48 hours ahead
l.run(duration=timedelta(hours=48))
# Print and plot results
print l
l.animation(filename='leeway_example.mp4')
l.plot(filename='leeway_example.png')
```

The final plotting command yields Fig. 2. The coastline shown is from the GSHHS database (Wessel and Smith, 1996), which is the default option used to check stranding in OpenDrift. This
coastline is however interfaced to OpenDrift as a regular Reader (Sec 2.1), and can be replaced by any other reader providing the CF variable *land_binary_mask*. This allows performing simulations in narrow bays or lakes where even the full resolution GSHHS coastline is too coarse.



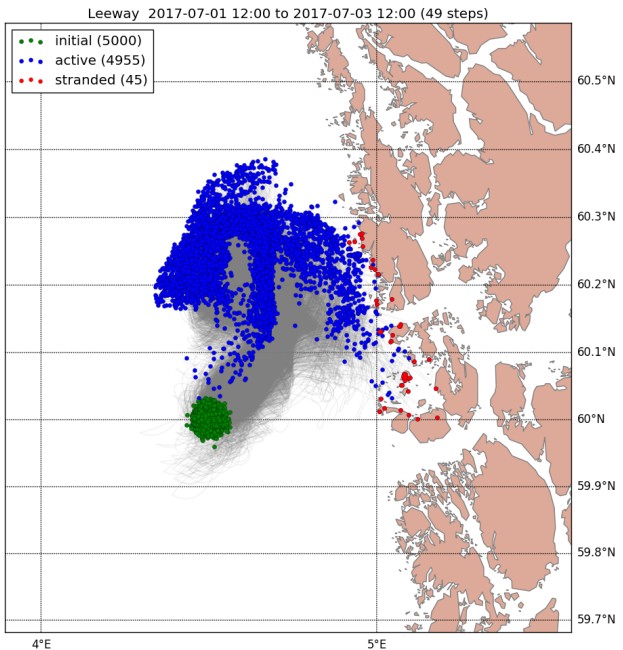

**Figure 2.** Output from the Leeway example of Sec 3.1. Green dots are the initial positions of the elements (life rafts), gray lines are trajectories, and blue dots are positions at the end of the simulation. Red dots indicate elements which have hit land (stranded).

## 3.2 OpenOil (Oil drift)

OpenOil is a full-fledged oil drift model, bundled within the OpenDrift framework. As a model it has been developed from scratch, but is based on a selection of parameterisations of oil drift as found in the open research literature. With regard to horizontal drift, three processes are considered:

– Any element, whether submerged or at the surface, drifts along with the ocean current.

– Elements are subject to Stokes drift corresponding to their actual depth. Surface Stokes drift is normally obtained from a wave model (or by any Reader), and its decline with depth is calculated as described in Breivik et al. (2016).

– Oil elements at the ocean surface are moved with an additional factor of 2% (configurable) of the wind. Together with the Stokes drift (typically 1.5% of the wind at the surface), this sums up to the commonly found empirical value of 3.5% of the wind (Schwartzberg, 1971). The physical mechanism behind this wind drift factor is not obvious, and is discussed in Jones et al. (2016).





The above three drift components may lead to a very strong gradient of drift magnitude and direction in the upper few meters of the ocean. For this reason, it is also of critical importance to have a good description of the vertical oil transport processes, which in OpenOil are the sum of the following factors:

– If the vertical ocean current velocity is available from a reader, the oil elements will follow it. This part of the movement is however often negligible compared to the processes below.

  – Oil elements at the surface, regarded as being in the state of an oil slick, may be entrained into the ocean by breaking waves. Presently OpenOil contains two different parameterisations of this entrainment rate, from which the user can chose as part of configuration (see below): 340   Tkalich and Chan (2002) and Li et al. (2017). The entrainment depends on both the wind and wave (breaking) conditions, but also on the oil properties, such as viscosity, density and oil-water interfacial tension.

  – Buoyancy of droplet is calculated according to empirical relationships and the Stokes law Tkalich and Chan (2002), dependent on ocean stratification (calculated from temperature and 345   salinity profile, normally read from an ocean model), oil and water viscosities and densities. Also, the buoyancy is strongly dependent on the oil droplet size (diameter), of which two parameterisations are available: one is a generic power law, with droplets between a minimum and maximum diameter, and a configurable exponent where -2.3 corresponds to the classical work of Delvigne and Sweeney (1989). The second option for droplet size spectrum is a 350   "modern" approach by Johansen et al. (2015), where a lognormal droplet spectrum is calculated explicitly based on wave height and oil properties such as viscosity, density, interfacial tension, and surface film thickness.

  – In addition to the wave induced entrainment, the oil elements are also subject to vertical turbulence throughout the water column, as parameterised with a binned-random-walk numerical 355   ical scheme described in Thygesen and Ådlandsvik (2007). This scheme is generic within OpenDrift, and is also used by the PelagicEgg module for ichtyoplankton (Sec 3.4). Only the properties specific to oil, or plankton, are coded in the respective classes (modules).

In addition to the vertical and horizontal drift, weathering of the oil also has to be considered. While parameterisations of weathering might also be implemented directly within the OpenDrift 360 framework, the OpenOil module instead interfaces with the already existing OilLibrary software developed by NOAA (https://github.com/NOAA-ORR-ERD/OilLibrary). The NOAA OilLibrary is also open source and written in Python, so integration is straightforward. In addition to state-of-the art parameterisations of processes such as evaporation, emulsification and dispersion, this software contains a database of measured properties of almost 1000 oil types from around the world. As oils 365 from different sources/wells have vastly different properties, such a database is of vital importance





for accurate results. The same OilLibrary is also used by the NOAA oil drift model PyGNOME (https://github.com/NOAA-ORR-ERD/PyGnome), where it is replacing the original ADIOS oil library (Lehr et al., 2002). PyGNOME includes also more processes not (yet) included in OpenOil, such as dissolution, and adding of dispersants.

To run an OpenOil simulation, one could re-use the exact code as for the Leeway example of Sec 3.1, only replacing the name of the imported module (*OpenOil* instead of *Leeway*), and replacing the *objectType* property of Leeway with a corresponding oil name from the NOAA database. However, whereas key features and functionality is shared among OpenDrift modules, each module (or group of modules) may add specific functionality. E.g., for OpenOil, it is possible to initialise

the simulations with an oil slick as read from a file containing contours, either a shapefile, or the GML-format/specification as used by the European Maritime Safety Agency (EMSA):

```
>>> o.seed_from_gml("RS2_20151116_002619_SCNB_HH_Oil.gml",
                    num_elements=2000)
>>> o.plot()
```

where o is the oil drift simulation object. The last command produces the plot shown in Fig. 3.

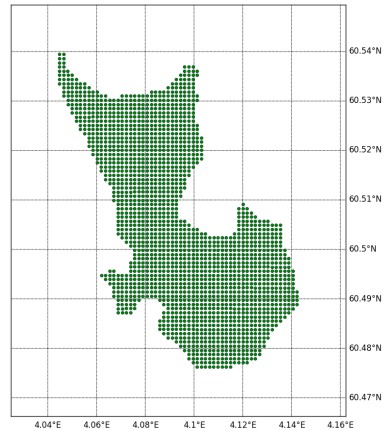

**Figure 3.** Oil drift simulation initialised by seeding 2000 oil elements within contours of an oil slick as observed from satellite (Radarsat2). The contour is imported from a GML file produced by Kongsberg Satellite Services (KSAT).

OpenOil also has module-specific configuration settings. The following commands specifies that the oil entrainment rate shall be calculated according to Li et al. (2017), and the oil droplet size spectrum shall be calculated according to Johansen et al. (2015).

```
>>> o.set_config('wave_entrainment:entrainment_rate', 'Li et al. (2017)')
```





```
>>> o.set_config('wave_entrainment:droplet_size_distribution',
                       'Johansen et al. (2015)')
```

Adjusting configuration this way is convenient for sensitivity studies, where one component is changed for two otherwise identical simulations.

After the simulation is finished, the generic plot command may be used to produce a map with
trajectories as shown in Fig. 2. However, more specific plotting methods are also available. The command `o.plot_oil_budget()` plots an oil budget as shown in Fig. 4.

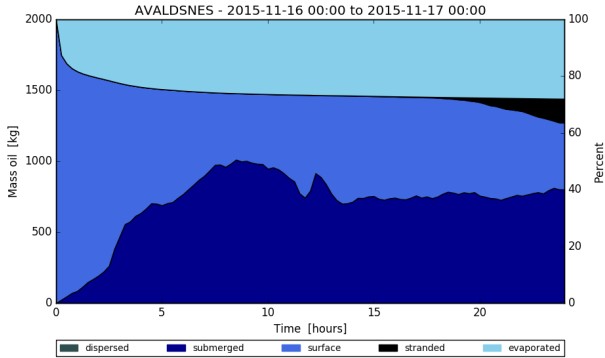

**Figure 4.** Plot of the time evolution of the oil budget of a 24 hour simulation with OpenOil. Of 2000 kg oil initially at the ocean surface, more than 20% is seen to evaporate within the first few hours. The amount of oil submerged due to wave action and ocean turbulence varies with the wind and wave conditions, with more oil resurfacing when wind decreases after about 8 hours. After about 18 hours, some of the oil seen to hit the coastline. These results are for the oiltype *"AVALDSNES"*, very different results could be obtained if using another oiltype for the same geophysical conditions.

The vertical distribution of the elements can be plotted with the command `o.plot_vertical_distribution()`, generating output as shown in Fig. 5. This method is shared among 3-dimensional modules, and may also be used for simulations with e.g. the PelagicEgg module (Sec 3.4).

**3.3  WindBlow (Atmospheric transport)**

As an example of a minimalistic trajectory model we also include the instance WindBlow, which simply calculates the propagation of a passive particle subject to a two-dimensional wind field. The code below is the complete, and fully functional WindDrift module.




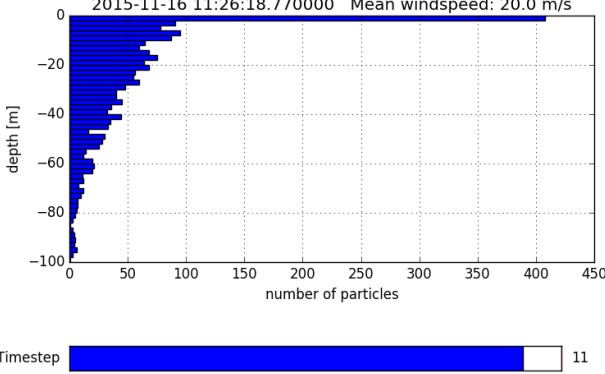

**Figure 5.** Vertical profile of the amount of (oil) elements. The bottom bar is an interactive slider, which the user can pull left/right to see the time variation of the vertical distribution.

```
# WindBlow module code
from opendrift.models.basemodel import OpenDriftSimulation
from opendrift.elements.passivetracer import PassiveTracer

class WindBlow(OpenDriftSimulation):
    ElementType = PassiveTracer
    required_variables = ['x_wind', 'y_wind']

    def update(self):
        self.update_positions(self.environment.x_wind,
                              self.environment.y_wind)
```

Because all common functionality is inherited from the main class, the WindBlow model only
needs to address its own specific needs: It will use elements without any other properties except for position (PassiveTracer), and the only forcing needed to move the elements is wind, whose vector components are named *x_wind* and *y_wind* in CF-terminology. The *update()* method which is called at each time step simply advects all elements with the wind velocity at their respective locations. The wind might be provided by any Reader (Sec 2.1). The WindDrift module may be run with an even
more simplified form of the code example found in Sec 3.1: the WindDrift class has to be imported, no Reader for ocean current is needed, and there is no object category to specify.




Clearly, an air parcel in the real atmosphere will also be subjected to updrafts and diffusion, and will with time rise or fall, but the example serves to demonstrate how little is required to develop a new subclass of OpenDrift. The model may be made more sophisticated by adding e.g. vertical wind (*upward_air_velocity*) and turbulence parameters to the list of required variables, and adding corresponding parameterisations of how to use this information for the advection.

### 3.4 Other modules

In addition to the models described above, some other modules are bundled within the OpenDrift repository, as illustrated in Fig. 6:

– **OceanDrift** is a basic module for tracking e.g. water masses or passive tracers. Stokes drift is included, if provided by a reader. A wind-drift-factor may also be specified, allowing an additional wind drag at the surface, e.g. for simulation trajectories of various ocean drifting buoys (Dagestad and Röhrs, 2017).

– **PelagicEgg** is a module for transport of pelagic ichtyoplankton. This module contains quite basic functionality with identical transport mechanisms as in Röhrs et al. (2014), including the vertical turbulent scheme (Sec 3.2) which is of key importance for most pelagic plankton applications. Although a fully working a module, users with specialised needs (e.g. a specific biological species) can customise the drift and behaviour parameterisations by modifying or adding parameterisations in the PelagicEgg module, such as larval behaviour. Some users have interfaced this module with existing Fortran-code for e.g. calculation of sunlight-dependent behaviour, see e.g. Kvile et al. (2017) and Sundby et al. (2017). A pure python version of the sunlight-module is available and will be included in a future version of OpenDrift.

– **ShipDrift** is a module for predicting the drift of ships larger than 30 meters, where the effect of waves has to be calculated explicitly, and not implicitly with the wind drift as in the Leeway module. This module is based on Sørgård and Vada (2011). A previous version programmed in the C programming language has been used operationally at MET Norway for 15 years, but is now replaced by the OpenDrift version, which has been tested and shown to provide identical output for identical input/forcing.

A module for drift of icebergs (OpenBerg, not yet included in repository) is being developed by Ron Saper at Carleton University with partial funding from ASL Environmental Sciences of Victoria, Canada, and with data support from the Canadian Ice Service (personal communication). Two different iceberg drift forecasting approaches are being tested. One approach uses a drag formulation to calculate wind and water drag forces. The challenge with this approach is that the trajectories are very sensitive to underwater draft, of which information is rarely available. The second approach predicts and subtracts the wind and tidal components of the drift, and then analyses the residual for extrapolation. Finally, wind and tidal components are added back in to produce a trajectory forecast.





Drift of marine plastics, including microplastics, is an important application not covered by modules included with the OpenDrift repository version 1.0. However, as most of the needed infrastructure is already provided, including a vertical mixing scheme, a user with knowledge of the relevant physics and basic Python programming should be able to implement such a module with moderate efforts. Though, there is no upper limit to the complexity of any module.

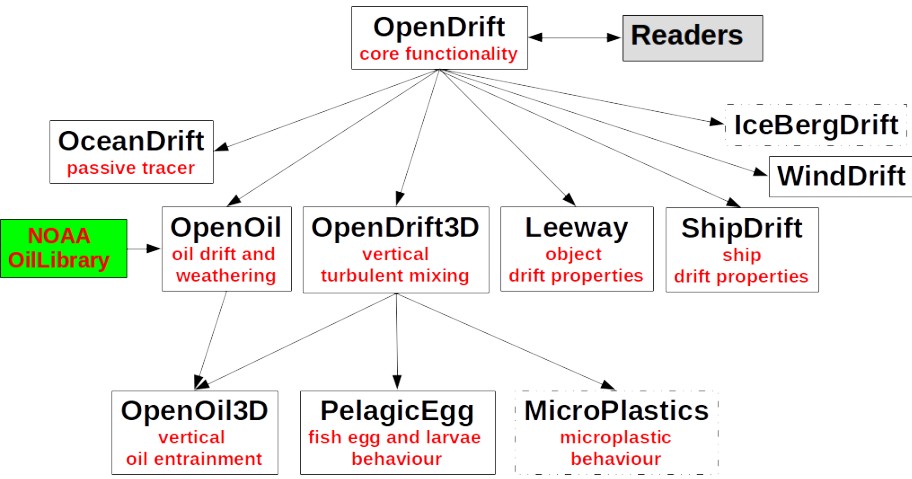

**Figure 6.** Illustration of how OpenDrift modules for specific applications (white boxes) inherit common functionality from the core module. This includes functionality to interact with Readers for obtaining forcing data. Sub-classing (inheritance) allows e.g. both the OpenOil and PelagicEgg models to share further 3D-functionality through subclassing the OpenDrift3D class. The boxes with solid boundaries illustrate existing modules bundled with the OpenDrift repository, whereas dashed boundaries indicate planned modules. The green box illustrates that OpenOil (oil drift model) utilises functionality from a third-party library, the NOAA OilLibrary.

## 4 Test suite and example scripts

OpenDrift contains a broad set of automatic tests (Python unittest framework) which can be run by the user to assure that all calculations are performed correctly on the local machine. The tests cover both basic calculations, such as interpolation and rotation of vectors from one spatial reference system (SRS) to another, but also more extensive integration tests, performing full simulations with the modules to verify that an expected numerical result is obtained. Also, very importantly, the tests are also run automatically on a variety of machine configurations, using the Travis Continuous Integration (CI) framework (https://travis-ci.org). This ensures that OpenDrift calculations remain accurate and correct with both old and new versions of the various required libraries (e.g. NumPy), and that



existing functionality is not broken as new functionality is added. For version 1.0 of OpenDrift, 64%
of the code is covered by the unit tests, as reported by the Coveralls tool (coveralls.io).

A user manual of OpenDrift is kept alongside of the code repository on the wiki-pages of GitHub
(https://github.com/OpenDrift/opendrift/wiki), facilitating a dynamic description to evolve with the
code, instead of diverting from it. Many example scripts (40 in version 1.0) are also provided in the
repository along with the needed input forcing data, illustrating a variety of real-life use cases. The
examples can easily be modified and adapted, allowing a soft learning curve.

OpenDrift also comes with a set of handy command line tools, such as *readerinfo.py*, which may
be used to easily inspect the contents and coverage of potential forcing fields. The following shell
command produces the same output as the example of Sec 2.1, where the switch '-p' also displays a
plot of the geographical coverage:

```
$ readerinfo.py ocean_model_output.nc -p
```

### 5   Graphical user interfaces

Although running OpenDrift modules with Python scripts (see e.g. Sec 3.1) is the most flexible and
powerful, a basic graphical user interface (GUI) is also included in the repository. A screenshot is
shown in Fig. 7. The GUI allows to select a module, and an object type or medium (e.g. oil type) cor-
responding to the module, and then a seeding location and time. The simulation is started by clicking
the "START"-button, and plots and animation of the output is available after the simulation, and also
saved to a NetCDF-file. The GUI will obtain forcing data through a provided list (configurable) of
Thredds-servers with global coverage, so there is no need for the user to obtain and download large
amounts of model input in advance. Although presently with only basic functionality, the GUI is in
operational use at MET Norway, where it is tested daily by meteorologist on duty as part of the oil
spill and search and rescue preparedness system.

In addition to the native GUI, a web interface has also been developed for remote access without
need for any local installation. This is based on communication with OpenDrift through a Web Pro-
cessing Service (WPS) developed at MET Norway (not included in the repository). Independently,
a WPS for the Leeway module has also been developed and implemented at the Swedish Met Office
(SMHI). A generic and configurable WPS to be included in the repository is planned for the future.





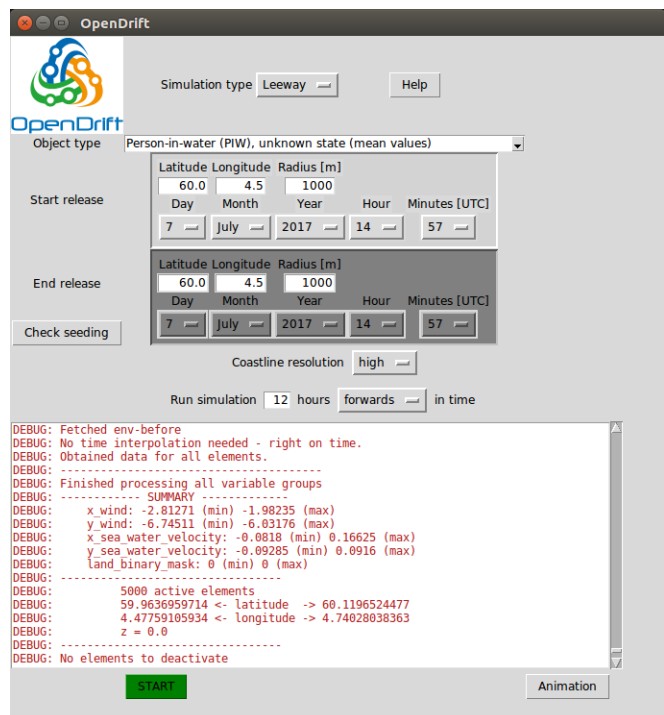

**Figure 7.** Screenshot of the Graphical User Interface included with OpenDrift.

## 6    Discussion and conclusions

Several offline trajectory models exist to predict the transport and transformation of various sub-
stances and objects in the ocean or in the atmosphere. OpenDrift is an open source Python framework
aiming at extracting anything which is common to all such trajectory models in a core library, and to
combine this with a clean and generic interface for describing any processes which are application-
specific. Several examples of such specific modules are bundled with the OpenDrift code repository,

and serve as ready-to-use trajectory models. This includes an oil drift model (OpenOil), a search
and rescue model (Leeway), and a model for predicting the drift and transformation of ichtyoplank-
ton (PelagicEgg). Interfaces ("Readers") towards the most common formats of forcing data (e.g.
NetCDF and GRIB) are also included, allowing any of the modules to be forced by data from a
combination of files and other sources, including remote Thredds servers. The concept of "Readers"

is also modularised, allowing a scientist or programmer to easily develop an interface towards any
other specific source of forcing data, such as e.g. an ASCII file containing *in situ* observations from
a buoy or weather station, or ocean currents from HF-radar systems in a specific binary format.





A built-in configuration mechanism provides flexibility to the operation of the OpenDrift modules. However, the fact that the application-specific processes of these modules are separated from the technical complexities of the OpenDrift core, provides even greater flexibility to the user in that it is easy to modify existing modules, or even write new modules from scratch. Several users have already developed or adjusted modules for their specific purpose, and added useful contributions to the OpenDrift core (Sundby et al., 2017; Kvile et al., 2017).

Whereas flexibility is important for scientific studies, OpenDrift is also designed for performance and robustness, and is in daily use for operational emergency response systems at the Norwegian Meteorological Institute. Being able to use the same tool in both cases, facilitates rapid transition of the latest research results into operations.

Another great benefit of the modularity provided by OpenDrift, is the ability to perform sensitivity tests by varying one component while keeping everything else constant. Much can be learnt from performing two otherwise identical simulations with e.g. input from two different Eulerian models, or by using two different parameterisations of some process. Further, consistency is also provided by the possibility of handling e.g. overlap of fish eggs and oil with the same forcing and numerical scheme. Traditionally, it might be difficult to draw conclusions by comparing the output from different trajectory models, as the differences depend on many factors, such as interpolation schemes and numerical algorithms.

The modules presently included with OpenDrift will be improved in the future, in particular by validation against available relevant observations. Among the general problems which require more attention, are to properly describe and quantify the very strong vertical gradients of horisontal drift often found in the upper few meters of the ocean, as result of a delicate balance between ocean currents and Stokes drift, as well as the direct wind drift affecting objects and substances at the very ocean surface. This vertical gradient of forcing is highly important for drift of e.g. oil and chemicals, plankton, and microplastics. This implies further that having accurate parameterisations of the vertical transport processes (wave entrainment, buoyancy and ocean turbulence) is also very important.

It is clear that waves modify the turbulent profile, and several recent studies (Belcher et al., 2012; Röhrs et al., 2012; Breivik et al., 2015) have demonstrated that this modifies the water properties, current profiles, and particle transport in the mixed layer. This has consequences for water level (Staneva et al., 2017) and ocean temperature (Alari et al., 2016), but more importantly for our work here, it is also likely to affect how deep small particles are mixed. There is ongoing work to model more realistically how suspended matter is moved vertically with changes in the level of turbulence. For buoyant particles such as cod eggs that remain in the ocean for long periods (Röhrs et al., 2014), this is absolutely essential to determining the trajectories since surface currents under the influence of wind and waves can differ greatly from subsurface currents.



To advance the understanding of transport processes in the upper ocean, more carefully controlled
experiments with high quality *in situ* measurements in the real ocean (and lower atmosphere) are
needed, to complement more abundant laboratory measurements. To maximise the outcome from
expensive measurements, it is also important to exchange knowledge across the various application
fields. E.g. a key for successful simulation of the drift of observed oil slicks in Jones et al. (2016)
was to incorporate a vertical mixing scheme developed for fish eggs (Sundby, 1983; Thygesen and
Ådlandsvik, 2007; Ådlandsvik and Sundby, 1994) into the oil drift model OpenOil.

## 7  Code availability

OpenDrift is housed on Github: https://github.com/OpenDrift/opendrift. The accompanying wiki
pages contain installation instructions, documentation and examples. Version 1.0 of OpenDrift is
registered with Zenodo: http://doi.org/10.5281/zenodo.845813. OpenDrift has been tested on both
Linux, Mac and Windows platforms. Version 1.0 requires Python 2.7, and is not adapted for Python
3. The OpenDrift framework is distributed under a GPLv2 license.

*Acknowledgements.*  K-FD, JR and ØB gratefully acknowledge support from the Joint Rescue Co-ordination
Centres through the project OpenLeeway and the Research Council of Norway through the CIRFA (grant no
237906) and RETROSPECT (grant no 244262) projects. The Norwegian Clean Seas Association (NOFO) and
the Norwegian Coastal Administration have been instrumental in their support and testing of the software during
the development phase.



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
