# Peer review of "OpenDrift v1.0: a generic framework for trajectory modeling"

_Geoscientific Model Development, 2017_

## Referee Comment (RC1) · Anonymous Referee #1 · 21 Sep 2017

In this manuscript, the authors present an overview of the design philosophy and some of the use cases for the new OpenDrift v1.0 particle tracking package. They highlight its flexibility and extendability to real-world problems.

This is a very readable, nicely-written manuscript describing an impressive piece of code. I have read it with much excitement, and congratulate the authors for some of the sophisticated software engineering they've done. They have seemingly mastered to make a particle tracking framework that is both very powerful as well as easy-to-use; not an easy achievement!

While I think this manuscript and the associated code is almost in a shape suitable for publication in GMD, I do have a few minor comments that the authors may want to address

[Figure]

- There is little actual benchmarking of the performance of the code. While the authors state that their code is efficient and fast, it would be good to see some quantification of these statements, ideally in comparison to similar other tools available. The advantage of providing benchmarks in this v1.0 paper is that it will also provide the authors a target for future versions of the code

- In general, I would have liked to see some more discussion of technical details and choices. This manuscript will probably become a GMD paper, with a readership of scientific code developers; yet the manuscript reads mostly like a high-level user guide. While I think most of these high-level descriptions are useful and should stay, a bit of 'under-the-hood' detail might please the technicallly-inclined reader

- On lines 6 and 7, is there a difference between 'model' and 'module'? If not, I suggest using the same word as it may be confusing to readers

- line 9: 'subjective estimates' here is a rather vague term. Perhaps explain a bit more what is meant?

- line 41: Another advantage of offline models, as the authors later also acknowledge, is their ability to track backwards in time

- line 61: 'robustness' for operational use is a rather vague term. What do the authors mean with this?

- line 70 and 76: The section header is named 'Reader class' but then later the authors state that Reader is a subclass. This may be confusing to some readers

- Table 1: How were these packages selected? Is this table really necessary? Some of the more widely used packages in large scale oceanography, such as Ariane and the Connectivity Modeling System, are missing

- line 82: Where is this 'ocean_model_output.nc' file located? It does not appear on the GitHub. Would it be good to provide access to these files somewhere, for users who want to recreate the simulation described here?

- line 83: Mention here already that OpenDrift supports 'lazy reading' of files? This is a really neat feature, as the authors also proudly acknowledge later, so best to at least mention it here too?

- line 127: Can any more technical details be provided of the caching mechanism? Does it use xarray? Dask? Something custom developed?

- lines 152-167: the frequent forward-referencing to later sections here makes for difficult reading. Can't the order of the sections be reshuffled somewhat to reduce the amount of forward-references?

- line 160: I am surprised that all history of particles is stored in arrays. Would these arrays not get potentially enormous? Are they kept in memory? Is there any smart memory-management implemented? What does this mean for scalability of the code to very many and/or very long trajectories?

- line 190: All examples of Reader subclasses have only one file. What if the data is in multiple files? Would that be a common use-case?

- line 225: How is 'evenly' defined here? Which algorithm is used?

- line 228: Is it also possible to create one's own seed_elements methods? Can they be overloaded?

- line 236: I assume 'runge-kutta' is a 4th order Runge-Kutta integration? Or 2nd order?

- line 252: Is the 'advective increments' the same as the time_step argument?

- section 3.1: some of this discussion, especially in the second paragraph, feels a bit out of place in a GMD paper about methodology and code. Is this really all necessary?

- line 287: some words seem missing at the end of this sentence

- line 314: Is stranding the only option available in OpenDrift? Or can one also use e.g. reflective boundary conditions?

[Figure]

- lines 516-540: again, much of this discussion does not seem very pertinent to the code and out of place in a GMD paper.

- There is no mention of paralel capacity for OpenDrift. Is the code parallelised, i.e. through numba? If not, are there any plans for a paralel version?

---

## Referee Comment (RC2) · Anonymous Referee #2 · 21 Jan 2018

General Comments

The authors present clear description of a new framework for Lagrangian particle modelling, with an emphasis on flexibility and wide scope for ongoing development. The review of literature (Sect. 1.1) is thorough and wide ranging. Justification for the new development (OpenDrift v1.0) is clearly articulated throughout the mansucript. The framework is clearly described (Sect. 2) and various case studies are briefly and clearly presented (Sect. 3). Overall, the manuscript is very well-written, with clear tables and figures throughout. It should be suitable for publication in GMD, subject to minor revisions in response to comments below.

Specific Comments

[Figure]

1. p.2, lines 42-46: It is true that the widely used ARIANE trajectory code is rather specifically build around the NEMO family of models. It may be work cited this example here.

2. p.3, end Sect. 1: It would be helpful to conclude the Introduction with a short paragraph outlining the rest of the paper, as is customary.

3. p.4, Table 1: Missing from Table 1 are ARIANE and CMS, the two particle trajectory codes perhaps most widely used by the marine science community.

4. p.17, Figure 4: Can you elaborate, in figure caption or text, what you define as "dispersed"; also, it is not really clear in the figure the distinction between "surface" and "submerged" – this is actually clearer in the figures posted at https://github.com/OpenDrift/opendrift/wiki/Gallery:-OpenOil-(oil-drift-model), where you define only four cases, with "in water" presumably "surface + submerged", which may serve as a better illustration in Sect. 3.2.

5. p.19, lines 434-441: What about the thermodynamics of the finite volume particles that represent the icebergs? Existing ocean-iceberg models take account of changes in iceberg dynamics (wind and water drag forces, also Coriolis and pressure gradient forces) due to changes in dimensions. Melting may not alter volume (and hence dynamics) on short operational timescales (hour-days), but will matter more on longer timescales (weeks-months). It would be useful to elaborate here on the extent of iceberg tracking (and accompanying physics/dynamics) that is currently planned.

---

## Author Comment (AC1) · 6 Feb 2018

General comments: In this manuscript, the authors present an overview of the design philosophy and some of the use cases for the new OpenDrift v1.0 particle tracking package. They highlight its flexibility and extendability to real-world problems. This is a very readable, nicely-written manuscript describing an impressive piece of code. I have read it with much excitement, and congratulate the authors for some of the sophisticated software engineering they've done. They have seemingly mastered to make a particle tracking framework that is both very powerful as well as easy-to-use; not an easy achievement! While I think this manuscript and the associated code is

almost in a shape suitable for publication in GMD, I do have a few minor comments that the authors may want to address

Comment 1.1. There is little actual benchmarking of the performance of the code. While the authors state that their code is efficient and fast, it would be good to see some quantification of these statements, ideally in comparison to similar other tools available. The advantage of providing benchmarks in this v1.0 paper is that it will also provide the authors a target for future versions of the code

Response 1.1. A paragraph has been added to the discussion section, illustrating typical calculation times for different types of simulations. A useful benchmark test for comparison with other models or future versions of OpenDrift would be hard to make, as the performance would depend on both the complexity of the calculations involved, as well as hardware configuration such as the speed of the local hard disks or network.

C 1.2, In general, I would have liked to see some more discussion of technical details and choices. This manuscript will probably become a GMD paper, with a readership of scientific code developers; yet the manuscript reads mostly like a high-level user guide. While I think most of these high-level descriptions are useful and should stay, a bit of 'under-the-hood' detail might please the technicallly-inclined reader

R 1.2. Some more technical details are added in Section 2.1 (Readers) about interpolation, and calculation of geophysical parameters from others, as well as about the newly added functionality of reading ensemble data. Some technical notes on performance is added to Section 6 (Discussion).

C 1.3. On lines 6 and 7, is there a difference between 'model' and 'module'? If not, I suggest using the same word as it may be confusing to readers

R 1.3. We are now consistently using the term "module" here.

C 1.4 line 9: 'subjective estimates' here is a rather vague term. Perhaps explain a bit more what is meant?
[Figure]

R 1.4. We have here replaced "subjective estimates" with "prescribed values".

C 1.5. line 41: Another advantage of offline models, as the authors later also acknowledge, is their ability to track backwards in time

R 1.5. This is a good point, which has now been added to the text.

C 1.6. line 61: 'robustness' for operational use is a rather vague term. What do the authors mean with this?

R 1.6. We believe this is quite well explained in the last paragraph of section 2.3.2: "A key feature of OpenDrift, for both convenience and robustness, is the possibility to provide a priority list of reader for a given set of variables. As an example it is possible to specify that a high resolution ocean model shall be used whenever particles are within coverage in space and time, and reverting to using another model with larger coverage in space and time whenever particles are outside the time- or spatial domain of the high resolution model. As an important feature for operational setups, the backup readers will also be used if the first choice model (file or URL) should not be available, or if there should be any other problems, such as e.g. corrupt values or files."

C 1.7. line 70 and 76: The section header is named 'Reader class' but then later the authors state that Reader is a subclass. This may be confusing to some readers

R 1.7. Reader is both a class, and at the same time a subclass of BaseReader. Thus we suggest to keep text as it is.

C 1.8 Table 1: How were these packages selected? Is this table really necessary? Some of the more widely used packages in large scale oceanography, such as Ariane and the Connectivity Modeling System, are missing

R 1.8. We agree that the table is not strictly necessary, but we would still prefer to keep it as a convenient overview of many available alternative models (yet stated to not be exhaustive). Ariane and CMS are obvious oversights, and these have been added to the table.

C 1.9. line 82: Where is this 'ocean_model_output.nc' file located? It does not appear on the GitHub. Would it be good to provide access to these files somewhere, for users who want to recreate the simulation described here?

R 1.9. This is a good suggestion. However, the given file is "well hidden" within the repository and has a long filename, which would draw a bit focus away from the fact that this is a simple illustration. Renaming the file would also have undesired consequences, thus we prefer to let the GitHub wiki and included example scripts serve as step-by-step tutorials.

C 1.10. line 83: Mention here already that OpenDrift supports 'lazy reading' of files? This is a really neat feature, as the authors also proudly acknowledge later, so best to at least mention it here too?

R 1.10. Yes, we fully agree that this is a key feature which can well be mentioned twice!

C 1.11. line 127: Can any more technical details be provided of the caching mechanism? Does it use xarray? Dask? Something custom developed?

R 1.1. The caching mechanism is custom developed, and so we have updated the text to read: "An internal caching mechanism is implemented". Discussion on how this is done would be too technical, and take focus away from presenting the main characteristics. We will, however, consider adding this discussion to the wiki/user manual.

C 1.12. lines 152-167: the frequent forward-referencing to later sections here makes for difficult reading. Can't the order of the sections be reshuffled somewhat to reduce the amount of forward-references?

R 1.12. Such reordering would then mean that details would be given before the main overview. Thus we would prefer to give the overview first as here, despite the need for forward references.

C 1.13. line 160: I am surprised that all history of particles is stored in arrays. Would these arrays not get potentially enormous? Are they kept in memory? Is there any

smart memory-management implemented? What does this mean for scalability of the code to very many and/or very long trajectories?

R 1.13. Yes, smart memory-management is implemented, but has not been mentioned. The following sentence is now added to section 3.6.2 (Exporting model output): "If the number of elements and time steps are too large to keep all data in physical memory, OpenDrift will flush history data to the output file as needed during the simulation to free internal memory."

C 1.14. line 190: All examples of Reader subclasses have only one file. What if the data is in multiple files? Would that be a common use-case?

R 1.14. Yes, this is in fact quite common, and we have now added the following sentence to illustrate how this works: "For netCDF files, it is also possible to create a single reader object which merges together many files, by using wildcards (* or ?) in the filename. This functionality is based on the NetCDF MFDataset class."

C 1.15. line 225: How is 'evenly' defined here? Which algorithm is used? R 1.15. We have replaced "evenly" with "with regular spacing". The algorithm calculates the area of the polygon, and calculates the x- and y-spacing needed to place the desired amount of elements within the polygon(s).

C 1.16. line 228: Is it also possible to create one's own seed_elements methods? Can they be overloaded?

R 1.16. Yes, this is possible. We have added the following sentence: "The seeding methods may also be overloaded to provide customised functionality for a given module."

C 1.17. line 236: I assume 'runge-kutta' is a 4th order Runge-Kutta integration? Or 2nd order?

R 1.17. We have now stated explicitly that this is a 2nd order scheme. As calculations are very fast (compared to file-reading), this allows using small time steps (e.g. 15

minutes), which is better for accurate checking of land-interaction, and which also leads to negligible difference of using Euler scheme or Runge-Kutta 2nd order. However, a 4th order scheme may still be implemented later.

C 1.18. line 252: Is the 'advective increments' the same as the time_step argument?

R 1.18. Spatial increment (i.e. delta_x, delta_y, delta_z) is meant here, but the same applies to the time increment.

C 1.19. section 3.1: some of this discussion, especially in the second paragraph, feels a bit out of place in a GMD paper about methodology and code. Is this really all necessary?

R 1.19. We agree that this may be shortened. The second paragraph has now been deleted.

C 1.20. line 287: some words seem missing at the end of this sentence

R 1.20. This was part of the paragraph which is now deleted.

C 1.21. line 314: Is stranding the only option available in OpenDrift? Or can one also use e.g. reflective boundary conditions?

R 1.21. Yes, there are three possible options, which is now described in Sect. 2.3.4 (Configuration).

C 1.22. lines 516-540: again, much of this discussion does not seem very pertinent to the code and out of place in a GMD paper.

R 1.22. We have deleted the lines 525-537, and merged the remaining text.

C 1.23. There is no mention of paralel capacity for OpenDrift. Is the code parallelised, i.e. through numba? If not, are there any plans for a paralel version?

R 1.23. There is presently no parallelisation, but this will be considered in the future. As reading of forcing data is often the bottleneck, parallel reading of input data (e.g.

currents and waves at the same time) might provide some benefit.

Anonymous Referee #2

General Comments: The authors present clear description of a new framework for Lagrangian particle modelling, with an emphasis on flexibility and wide scope for ongoing development. The review of literature (Sect. 1.1) is thorough and wide ranging. Justification for the new development (OpenDrift v1.0) is clearly articulated throughout the mansucript. The framework is clearly described (Sect. 2) and various case studies are briefly and clearly presented (Sect. 3). Overall, the manuscript is very well-written, with clear tables and figures throughout. It should be suitable for publication in GMD, subject to minor revisions in response to comments below.

Comment 2.1. p.2, lines 42-46: It is true that the widely used ARIANE trajectory code is rather specifically build around the NEMO family of models. It may be work cited this example here.

Response 2.1. According to the CMS website, it is used with forcing from different models, including e.g. NEMO and HYCOM. Thus we would like to be careful to mention specific examples of trajectory models which are tied to an ocean model, as the code might have been generalised in the meantime.

C 2.2. p.3, end Sect. 1: It would be helpful to conclude the Introduction with a short paragraph outlining the rest of the paper, as is customary.

R 2.2. An outline is now presented at the end of Sect 1.

C 2.3. p.4, Table 1: Missing from Table 1 are ARIANE and CMS, the two particle trajectory codes perhaps most widely used by the marine science community.

R 2.3. References have also been added to Ariane and CMS models. Yes, these were major oversights.

C 2.4. p.17, Figure 4: Can you elaborate, in figure caption or text, what you

define as "dispersed"; also, it is not really clear in the figure the distinction between "surface" and "submerged" – this is actually clearer in the figures posted at https://github.com/OpenDrift/opendrift/wiki/Gallery:-OpenOil-(oil-drift-model), where you define only four cases, with "in water" presumably "surface + submerged", which may serve as a better illustration in Sect. 3.2.

R 2.4. It was a bit misleading that the category "dispersed" was included in the legend, although there was no dispersion (removal of very small oil particles) for that example. The code has now been updated so that only categories which contribute to the oil budget are shown in the legend, and the figure has been replaced.

C 2.5. p.19, lines 434-441: What about the thermodynamics of the finite volume particles that represent the icebergs? Existing ocean-iceberg models take account of changes in iceberg dynamics (wind and water drag forces, also Coriolis and pressure gradient forces) due to changes in dimensions. Melting may not alter volume (and hence dynamics) on short operational timescales (hour-days), but will matter more on longer timescales (weeks-months). It would be useful to elaborate here on the extent of iceberg tracking (and accompanying physics/dynamics) that is currently planned.

R 2.5. Thermodynamics is not included in the first version of OpenBerg. This is now mentioned in the text.

NOTE: An important new feature since the first submission of this manuscript, is that OpenDrift now may read forcing data from ensemble models, and use different ensemble members for the different elements. A paragraph about this is added, in lines 153-159.